# Validation of a Real-Time PCR Assay for Identification of Fresh and Processed *Carica papaya* Botanical Material: Using Synthetic DNA to Supplement Specificity Evaluation

**DOI:** 10.3390/foods12030530

**Published:** 2023-01-25

**Authors:** Rajesh Patel, Adam C. Faller, Tiffany Nguyen, Zheng Quan, Corey Eminger, Swetha Kaul, Ted Collins, Yanjun Zhang, Peter Chang, Gary Swanson, Zhengfei Lu

**Affiliations:** 1Quality Control Laboratory, Herbalife International of America, Inc.—Winston-Salem, 3200 Temple School Road, Winston Salem, NC 27107, USA; 2Corporate Quality Laboratory, Herbalife International of America, Inc., 950W 190th Street, Torrance, CA 90502, USA; 3Corporate Quality, Herbalife International of America, Inc., 990W 190th Street, Torrance, CA 90502, USA

**Keywords:** papaya authentication, botanical identification, genomic authentication, synthetic DNA, specificity evaluation

## Abstract

Several commercially important botanicals have a lack of diagnostic testing options that can quickly and unambiguously identify materials of different matrices. Real-time PCR can be a useful, orthogonal approach to identification for its exceptional specificity and sensitivity. *Carica papaya* L. is a species with a lack of available identification methods, and one which features two distinct commercially relevant matrices: fresh fruit and powdered fruit extract. In this study, we demonstrate the successful design and validation of a real-time PCR assay for detection of papaya DNA extracted from the two matrices. We also propose a technique that can be used during exclusivity panel construction, when genuine botanical samples are not available for certain species: substitution with synthetic DNA. We demonstrate the use of this material to complete a comprehensive specificity evaluation and confidently determine suitable C_t_ cutoff values. Further, we demonstrate how ddPCR can be used to determine the copy number of the target sequence in a set amount of genomic DNA, to which synthetic DNA samples can be corrected, and how it can verify specificity of the primers and probe. Through the presentation of successful assay validation for papaya detection, this work serves as a guideline for how to approach specificity evaluation when non-target botanical samples are difficult to obtain and otherwise may not have been included in the exclusivity panel.

## 1. Introduction

Papaya (*Carica papaya* L.) is a globally important botanical, recognized as a food and nutraceutical. Consumption of the fruit is common, fresh or as a processed extract, but all parts of the papaya plant are used medicinally, including the root, bark, seed, peel, and pulp [1,2]. Papaya fruit, which contains a high level of antioxidants with a low calorie count, has been used to reduce the risk of heart disease, improve blood sugar control in people with diabetes, and aid in weight management [3]. Several bioactive compounds isolated from this botanical have been studied and associated with potential health benefits, including the chymopapain and papain enzymes, thought to aid in digestion, and relief of arthritis [4]. The papaya market is consistently growing, with a projected CAGR of 3.29% from 2022 to 2026 (~2.5 million metric tonnes of increased production) [5]. In 2010, global papaya production reached 11.22 million tonnes, and 14.1 million tonnes in 2020 [6,7]. The rapid increase in production is driven by consumer demand and influenced by the expanding range of cultivation area past the typical tropical and subtropical regions [8]. Climate change and advanced farming techniques have more recently allowed for the cultivation of papaya in Mediterranean countries, such as greenhouses in Sicily [8,9]. In any growing market part of the food industry, businesses must be wary of economically motivated adulteration. Vulnerabilities can be exploited at many stages of the supply chain, allowing unscrupulous industry players to introduce or trade inauthentic materials [10]. For this reason, effective analytical methods in established quality control programs must be able to verify material identity and potency. In addition to the importance of paying consumers deserving authentic material, therapeutic efficiency of drugs in traditional medicine depends on the use of genuine raw materials [4].

The FDA’s dietary supplement cGMPs (current Good Manufacture Practices) require supplement manufacturers to confirm the identity of components before use, by appropriate, scientifically valid methods (21 CFR 111.75) [11]. Performing identity tests on botanical materials can be challenging due to the complex phylogeny of closely-related species and variations in phytochemical profiles [12]. Matrix or botanical-specific challenges can impact one method of testing over another; thus, orthogonality of testing can provide a solution [13]. For example, grinding material eliminates many indicative morphological characteristics, precluding visual identification; chemical variation may make development of standardized monographs difficult; and degraded DNA in sterilized powders and extracts may result in failed PCR reactions [12]. Each of these challenges may be overcome by employing a combination of analytical techniques where they are most appropriate, hence making use of the fundamental taxonomic determination of morphology, the consistency in measurement of chemical analytes, and the species-level identification of genomic material derived from any plant part [12]. In this study, papaya (*Carica papaya*) fruit and a fruit powder extract (8–10%) were used in the development and validation of a qualitative, real-time PCR assay. Although there have been many nutritional analyses on papaya fruit, the identification of highly processed papaya fruit powder has not been heavily investigated [2]. DNA analysis of this botanical has been relegated to methods such as RAPD, which may not be suitable for herbal product authentication due to a lack of reproducibility [14,15,16,17]. Dhanya et al. developed a more reliable SCAR marker in 2009, but only in the context of differentiating papaya from black pepper [18]. Chemical marker compounds for papaya fruit are not often reported, except for enzymes or other non-specific water-soluble components, which are not ideal targets for conventional chemical analytical methods like HPLC [19]. There has been limited exploration into phytochemical fingerprinting, using techniques like GC-MS [15,20], NIR spectroscopy [21], and HPTLC [22]. These methods were able to discriminate papaya from non-target species, but authors noted disadvantages like time-consuming sample preparation and the requirement for skilled human resources [21]. These studies identified a need for development of more simple, efficient, and sensitive methods for papaya as a target analyte. Genomic methods should be further explored as orthogonal approaches to papaya detection, and real-time PCR assays are a good candidate for their accessible operation and result interpretation, once designed. In literature, scientists in the analytical testing field have recommended exploration of PCR and qPCR for food authentication, due to high sensitivity and specificity, fast turnaround time, and low cost [23]. To extend application to the identification of highly processed botanical materials, where extractable DNA can be degraded, species-specific PCR that is designed for small genomic targets (i.e., <200 bp) should be prioritized [24,25].

This study provides an alternative option for the identification of papaya materials, with the development of a papaya-specific, real-time PCR assay. Targets were determined in silico, based on characteristic DNA sequences in the *C. papaya* genome, and the assay was validated in the laboratory. The method was demonstrated to allow species-specific identification of the target (evaluated using an exclusivity panel of common botanical non-target reference materials), with superior PCR efficiency and sensitivity. Another main focus of this study dealt with the logistical challenges of obtaining raw materials for difficult-to-procure species that are part of the exclusivity panel, and the importance of finding methods that will allow for exhaustive specificity evaluation and determination of appropriate C_t_ cutoffs. In these situations, we explored the technique of including gBlocks^®^ (synthetic DNA) that represented the assay-relevant genomic regions of the non-targets for which we were unable to obtain raw botanical materials. This technique can aid in the creation of a comprehensive exclusivity panel, increasing confidence in the specificity of an assay. The absolute quantitation capability of ddPCR allowed for the concentration of gBlocks^®^ to be appropriately corrected for testing (to reflect the intended target-sequence copy number) and provided an interesting contrast to qPCR interpretation, since the same primers and probe were used. Using synthetic DNA for non-targets allowed for an evidence-based determination of C_t_ cutoffs, which should be chosen with care to minimize both type I and type II error [26]. Repeatability and reproducibility of the assay were evaluated, with the intention of demonstrating its capability as a routine quality-control tool for delivering same-day identification results.

## 2. Materials and Methods

### 2.1. Sample Collection

The *Carica papaya* real-time PCR assay was validated using market samples from three different countries of provenance, two different batches of powdered papaya extract (8–10% powder), and *Carica papaya* botanical reference materials. Prior to validation, the three papaya market samples (fresh fruit) were authenticated via Sanger sequencing on an Applied Biosystems^®^ 3500 Genetic Analyzer, using a BigDye™ direct cycle sequencing kit (Applied Biosystems, Waltham, MA). In addition to target papaya samples, a total of 21 non-targets botanical samples (closely related species, or species commonly used as ingredients or adulterants in food and dietary supplements) were collected to test the specificity of the assay. Authentic target and non-target botanical reference materials were obtained from sources including Chromadex, Alkemist, BI Neutraceuticals, Martin Bauer, and American Herbal Pharmacopoeia (AHP) (Chromadex, Los Angeles, CA, USA; Alkemist, Garden Grove, CA, USA; Martin Bauer, Secaucus, NJ, USA; American Herbal Pharmacopeia, Scotts Valley, CA, USA). In addition, two synthetic gBlocks^®^ were ordered from IDT (Integrated DNA Technologies, Coralville, IA, USA) to represent the target sequence of species for which botanical samples could not be procured. See Table 1 for the list of target and non-target samples used in the study.

### 2.2. Primer and Probe Design

Over 130 ITS2 barcode sequences of common botanical species used in food and dietary supplements were downloaded from Herbalife Botanical DNA Barcode Database (NCBI Bio Project: No. PRJNA503738) and imported into the R statistical software [27]. Primers were designed to specifically target the *C. papaya* species using the “Design Primers” R package, with default settings (https://rdrr.io/bioc/DECIPHER/man/DesignPrimers.html accessed on 1 April 2021). These primers follow typical design conventions, including similar melting temperatures (T_m_ ± 2 °C) between forward and reverse primers, 18–30 bp lengths, avoidance of mononucleotide guanine repeats (those with more than four bases), and 35–65% GC content [28,29]. The qPCR probes were also designed with several considerations, including the T_m_ of the probe being 4–6 °C higher than that of the primers, the oligo being 18–30 bp in length, annealing temperatures (T_a_) of oligos being ≤5 °C below T_m_, GC content being 40–60%, and the 5’ end of the probe not ending on a guanine [28,29]. Thermodynamic criteria should also be a consideration; strong internal hairpin structures and homodimers should be avoided (with ΔG < −9.0 Kcal/mole), as well as extendable heterodimer formation [30].

### 2.3. DNA Extraction and Quantification

Genomic DNA from samples were extracted using a DNeasy *mericon* food kit (Qiagen, Germantown, MD, USA), according to the manufacturer’s manual. For both raw and highly processed botanical materials, 50 to 55 mg of dry material was used as the input for extraction, and the final product was eluted in 50 µL of elution buffer. After extraction, DNA concentrations were measured with a Qubit^TM^ 4.0 fluorometer, using the associated Qubit^®^ dsDNA high-sensitivity assay kit (Invitrogen, Carlsbad, CA, USA).

### 2.4. Real-Time PCR

All PCR reagents and oligos were purchased from IDT. Following design and in silico specificity analysis, primer and probe sequences were synthesized as they appear in Table 2. Real-time PCR worked by hydrolysis-probe-based chemistry.

The *C. papaya* identification assay was performed in a 20 μL PCR mix, containing 10 μL Prime-Time Gene Expression Master Mix, 2 μL primer mix (of a 5 μM mixture of forward and reverse primers), 2 μL probe (2.5 μM), and 6 μL gDNA extracted from botanical materials (Master Mix: Integrated DNA Technologies, Coralville, IA, USA). 

The following thermocycling protocol was used for real-time PCR, on an Applied Biosystems™ 7500 Real-Time PCR System (Applied Biosystems, Waltham, MA): (1) an initial denaturation step of 3 min at 95 °C and (2) 50 cycles of 15 s at 95 °C and 30 s at 60 °C, with a fluorescent reading measured in the FAM channel after every cycle. A negative control of nuclease-free water was included in each run. ROX reference dye was added at a concentration of 20 µL in 5 mL of the master mix (2×) for experiments performed on the 7500 Real-Time PCR System.

The same primers and probes from the real-time PCR assay were used in all ddPCR experiments.

### 2.5. Assay Specificity

The specificity of an assay is determined by the set of oligonucleotides’ (i.e., primers and probe) reactivity with the target and non-target species. Non-targets may include closely related species (congeneric or confamilial) or other commercially relevant botanicals. Good specificity involves exclusive amplification of intended targets, and no amplification of non-targets. Specificity of the assay was tested following guidelines for validation of qualitative real-time PCR methods for identification of botanicals [31]. The specificity test results were conveyed as a percentage of false positives and negatives, both of which should be zero (see formulae):*TP* (%) = [(number of correctly classified known positive samples)/(total number of known positive samples)] × 100(1)
*FP* (%) = [(number of misclassified known negative samples)/(total number of known negative samples)] × 100(2)
*FN* (%) = [(number of misclassified known positive samples)/(total number of known positive samples)] × 100(3)
where: *TP* = true positive, *FP* = false positive, *FN* = false negative.

Specificity of the *C. papaya* assay was evaluated using six target samples (DNA extracted from four raw materials and two processed powder samples), along with 23 non-target species—including possible adulterants, confamilial species, and biological reference materials (BRMs) with taxonomic herbarium vouchers (Table 1). A no-template control (NTC) was included in all validation tests.

#### gBlocks^®^ for Unprocured Species and ddPCR Copy Number Determination

The two species that were closely related to papaya and unable to be procured as raw materials were instead represented by synthetic DNA gBlocks^®^. The gBlocks^®^ for *Jacaratia dolichaula* and *Jacaratia digitata* were used as a template in reactions. The *Jacaratia dolichaula* gBlock^®^ was designed based on a portion of the sequence from the JX092060.1 GenBank accession, and the *Jacaratia digitata* gBlock^®^ was designed based on a portion of the sequence from the MK914407.1 GenBank accession (Appendix A). The ddPCR was used to determine copy number of the assay target sequence in papaya DNA, based on a 0.25 ng/µL loading concentration. The resulting copy number was used as the benchmark to which the gBlock^®^ copy number could be corrected. Further, the copy number was also evaluated for low-input DNA that was extracted from processed material, in order to serve as a benchmark for the typical DNA yield from that type of matrix. Experiments were carried out on a QX200 Droplet Digital PCR System (using a PX1 plate sealer) (Bio-Rad, Irvine, CA, USA).

### 2.6. Assay Efficiency and Sensitivity

Assay efficiency is evaluated by creation of a standard curve consisting of five 10-fold dilutions of target DNA, each tested in triplicate (as recommended by the published assay design guidelines) [31,32]. This curve also allows for determination of linearity, expressed as a correlation coefficient (R^2^). Acceptable performance thresholds for this assay were set to R^2^ ≥ 0.98 (linearity) and 80–120% efficiency. In this case, two standard curves were generated, using DNA extracted from two different papaya seed sources (Brazil and USA), each at a starting concentration of 0.25 ng/µL. Consequently, these curves test the linear dynamic range of the assay, describing the upper and lower limits of detection. The parameter of sensitivity can be expressed as the lower detection limit (LOD) for amplification of the intended target. Here, the lowest serial dilution of the standard curve, where all three replicate samples are positive, serves as the LOD. To note, DNA inputs will be described as concentrations (e.g., 0.25 ng/µL), and associated quantity in reactions is based on a sample loading volume of 6 µL.

### 2.7. Assay Repeatability

This parameter is measured as a percent agreement of true positive and negative results obtained for replicated samples analyzed in the same laboratory, by the same operator, on the same device. Repeatability for the *C. papaya* assay was tested on three target samples and one non-target sample, all in triplicate, by the same operator, on the same device, on two different days. Acceptable repeatability for this assay was 100% true positives, and 0% false negatives and false positives.

### 2.8. Assay Reproducability

This parameter is measured as a percent agreement of true positive and negative results obtained for replicated samples analyzed in two laboratories, or by two operators. Reproducibility for the *C. papaya* assays was tested on three target samples, in triplicate, by two different operators, in two different labs. Acceptable reproducibility for this assay was 100% true positives, and 0% false negatives and false positives.

### 2.9. ddPCR Comparison

The ddPCR was also used to test target samples of DNA extracted from fresh papaya and powdered papaya extract, and the gBlock^®^ DNA of the non-target *Jacaratia dolichaula* and *Jacaratia digitata*. Two experiments were run, one to 40 PCR cycles and one to 50 PCR cycles, and all samples were tested in duplicate. All other thermocycling parameters were the same as real-time PCR.

## 3. Results

### 3.1. Authenticity Testing of Papaya Samples

The three *C. papaya* samples that were procured from local and global markets were authenticated using Sanger sequencing. Sequences obtained from Brazil, Guatemala, and USA-derived papaya samples were aligned to GenBank using NCBI Blast, and matched with 100% identity to *C. papaya* accessions. Sequences from an in-house voucher and AHP BRM were also aligned, matching 100% and 99.56%, respectively. Two ambiguous, uncalled bases represented the mismatches in the AHP BRM sequence. Figure 1 shows the ITS2 alignment for the *C. papaya* samples, and Appendix A shows the *rbcL* alignment.

### 3.2. Assay Specificity

Specificity of the *C. papaya* assay was experimentally assessed using target DNA extracted from botanical reference material and authenticated *C. papaya* fruit, along with an exclusivity panel of non-target species (Table 1). All target samples of fresh papaya showed positive amplification before 25 cycles. Target papaya samples that were highly processed powders (low-quantity and -quality DNA) amplified at a much later C_t_ (~30–35 cycles). To observe a full sigmoidal amplification curve for these samples, the PCR assay was performed to 50 cycles. While *C. papaya* DNA revealed positive amplification, no amplification curves were observed for any of the 21 non-targets species for which DNA was extracted from raw botanical materials (after 50 PCR cycles) (Figure 2). Since none of these non-targets amplified, the number of false positive cases is equal to zero, meaning the specificity of the assay is 100%. The remaining two non-targets, represented by gBlocks^®^, were tested in a subsequent run.

### 3.3. Assay Efficiency and Sensitivity

To determine the amplification efficiency of the *C. papaya* identification assay, DNA extracted from two different sources of papaya was serially diluted, 10-fold, from 0.25 ng/μL to 25 fg/μL (each dilution point tested in triplicate). As shown in Figure 3, the standard curve from the USA-sourced papaya revealed an assay efficiency of 104% and a linearity of R^2^ = 0.999. Amplification efficiency using the Brazil-sourced papaya was 105%, with an R^2^ of 0.998 (Appendix A). Both curves exhibited superior efficiency within the acceptable range (80–120%), and linearity over the acceptably threshold (R^2^ ≥ 0.98). At the most diluted data point, positive amplification of all replicates was observed (C_t_ values between 32.9–33.5), allowing determination of the assay LOD as 25 fg/μL. In addition, the upper end of the dynamic range was represented by the first data point in the dilution series (0.25 ng/µL) with C_t_ values between 19.21–19.56.

### 3.4. Assay Repeatability and Reproducability

Repeatability testing, which involved a two-day, repeated test of three *C. papaya* samples and one non-target (reactions in triplicate), revealed 100% true positives and 0% false negatives or false positives. The assay produced very similar C_t_ values for the three positive samples, across the two days (means of 19.4, 19.6, and 20.2 on the first day were comparable to means of 19.3, 20.4, and 19.5, respectively, on the second day) (Appendix A). The non-target sample (*Melissa officinalis*) did not amplify.

Similarly, when two different operators tested the assay, in different labs (reproducibility), using papaya DNA extracted from processed powder samples, C_t_ values of triplicate test results were all positive (means of 39.5, 36.17, and 33.67 from the first lab were comparable to means of 32.17, 32.97, and 33.7, respectively, from the second lab) (Appendix A). The non-target sample (*Melissa officinalis*) did not amplify. To note, DNA extracted from processed powder samples was not able to be quantified by the Qubit^TM^ 4.0 Fluorometer, due to the quantity of material being below the limit of quantification (LOQ) of the device. The DNA input from samples was kept consistent based on a 6 µL volume of DNA elute input in reactions.

### 3.5. gBlocks^®^ Used for Further Specificity Evaluation

The gBlocks^®^ for *Jacaratia dolichaula* and *Jacaratia digitata* were first corrected to a copy-number that corresponded to that of 0.25 ng/µL *C. papaya* genomic DNA (gDNA). This was achieved via ddPCR, which allowed for the determination of a copy-number equivalency of 28,800 copies/µL to 0.25 ng/µL of papaya DNA. In addition, six samples of DNA extracted from processed papaya material were run in a ddPCR experiment to determine the average copy-number yield in 6 µL of a DNA extract. This was meant to find a copy-number that reflects the typical DNA yield of an extraction from processed papaya powder. The average of six ddPCR readings was used (standardization based on 6 µL input into the reaction) because the concentration of the DNA extracts was unable to be read by the Qubit^TM^ 4.0 Fluorometer, since the quantities were below the LOQ of the instrument. An average of 7.21 copies/µL was determined and used as the input for a low-concentration specificity test.

At a high concentration (0.25 ng/µL; 28 800 copies/µL) (3 ng loaded based on 6µL), the papaya DNA amplified with an average C_t_ value of 17.23. At this copy number input, the *Jacaratia dolichaula* DNA was amplified with an average C_t_ of 32.54, and the *Jacaratia digitata* DNA was amplified with an average C_t_ of 37.11. At the low concentration input (7.21 copies/µL), the average C_t_ value for a 6 µL input was 32.77 for papaya DNA, and 42.60 for *Jacaratia dolichaula. Jacaratia digitata* DNA did not amplify at this input quantity.

To note, the non-targets were tested at the high- and low-input quantities (based on the target sequence copy-number), first with gBlock^®^ as the only genetic material template, and then with gBlock^®^ plus *Angelica sinensis* “carrier gDNA” (i.e., the addition of gDNA from an unrelated species mimics the typical environment of a target sequence in the presence of the rest of the genome, as opposed to the pure target sequence environment of the gBlock^®^). There was no discernable C_t_ difference between the gBlock^®^ alone and gBlock^®^ plus carrier gDNA samples.

### 3.6. ddPCR

In addition to using ddPCR to determine target sequence copy-number of papaya DNA, experiments were run to evaluate the use of ddPCR for detection of papaya targets. In both the 40 and 50 PCR cycle experiments, a high-fluorescent amplitude with 100% positive droplets were observed for fresh papaya samples. In these cases, a copy-number call was not able to be made due to the over-concentration of the target. The samples of DNA extracted from the papaya powder showed fewer positive droplets, but a clear clustering of droplets in a high-fluorescent amplitude (~1000). *Jacaratia digitata* samples did not show any clear amplification, and *Jacaratia dolichaula* samples showed a uniform scatter of positive droplets from low- to mid-fluorescent amplitude (from zero to ~800 for 40 cycles, and zero to ~900 for 50 cycles) (Figure 4).

## 4. Discussion

### 4.1. Assay Design and Performance

Rapid, real-time PCR detection methods for botanicals can be advantageous additions to quality-control programs if key performance features are met and the assay can accommodate relevant matrices. Exceptional specificity is paramount with this type of assay because in the absence of sequencing, a positive identification is based upon the selective and unambiguously identifiable amplification of the target species’ DNA. Specificity can only be properly evaluated with construction of a comprehensive inclusivity and exclusivity panel [31,33].

All primer and probe designs begin in silico, as did the design of this assay with the discovery of a papaya-specific sequence within the ITS2 genomic region. However, only through laboratory testing can the efficiency of amplification and the potential affinity of oligos for similar, non-target sequences, be evaluated. There are several factors in a reaction, untestable in silico, that can influence the thermodynamic interactions of oligos, such as the presence of inhibitory molecules of plant origin, or DNA degradation [13]. Papaya and many other botanicals include compounds like phenols, that are inhibitory to PCR, and concentration of these compounds can vary based on the matrix [34]. In this assay design, DNA extracts from both fresh papaya fruit and processed papaya powder were included as an inclusivity panel to evaluate amplification of the target sequence in different matrices. Papaya fruits of three different countries of provenance were collected, because location-specific climatic or edaphic influences on the chemical composition of plants can consequently determine PCR inhibition [12]. Positive amplification of all samples indicated acceptable efficiency and accuracy of the assay and demonstrated its tolerance for target DNA extracted from different matrices.

A common challenge when working with highly processed botanical material is the occurrence of DNA degradation. Many herbal extract products undergo several heat processes that can degrade DNA, thus negatively impacting the quantity and quality of the residual nucleic acid in the botanical material [35,36,37]. *C. papaya* 8–10% fruit powder extract undergoes pasteurization at 105 °C for 60 s, followed by spray-drying at 116 °C for ~5 min. Spray-drying occurs using maltodextrin on a glucose substrate. Literature indicates that DNA denatures at approximately 90 °C and will permanently degrade at temperatures of 130 °C or higher [38]. Permanent degradation can occur at temperatures below 130 °C when increased pressures are introduced [38]. As demonstrated, DNA degradation is a common factor for botanical extracts, thus it is important for an assay to be effective in identifying target DNA at very low concentrations. Positive amplification of DNA from all *C. papaya* powder samples suggests that the assay can be used for identification of processed material.

A comprehensive exclusivity panel should include relevant, closely related species and other commercially relevant botanicals, as to determine unambiguous identification of the target species [33]. Late amplification of non-targets may be permissible, so long as there is a clear separation between the non-target C_t_ and the lowest concentration of target DNA, and an appropriate C_t_ cutoff can be determined [26]. In this case, the 21 non-targets for which botanical samples were collected did not amplify (even after 50 PCR samples), based on 28,800 copies/µL (3 ng) of loaded DNA. In practice, botanical samples representative of all materials should be collected, but this task is not always straightforward to complete. Analytical authorities like AOAC offer guidelines to exclusivity panel sampling, suggesting that practically obtainable non-target botanicals be sourced [39]. In this study, we expand the interpretation of “practically obtainable” and demonstrate that in cases where materials for non-target species are difficult to procure, synthetic gBlock^®^ DNA can be used as a substitute for genomic DNA extracted from botanical samples. Two difficult-to-procure species closely related to papaya, *Jacaratia digitata* and *Jacaratia dolichaula*, were intended to be included in the exclusivity panel for specificity evaluation of the assay [40]. Two gBlocks^®^ were tested (representing non-target sequences homologous to the assay’s target papaya sequence), corrected for the copy number to reflect the same amount of target sequence as the papaya gDNA. These two species showed late amplification, meaning the gBlock^®^ technique was essential in determining appropriate C_t_ cutoffs for the assay if false positives from *Jacaratia digitata* or *Jacaratia dolichaula* are to be avoided. For each of the two gBlocks^®^, two different input levels were tested. The high copy number input (28,800 copies/µL) was intended to reflect the input amount if using DNA extracted from fresh papaya. Since the *Jacaratia digitata* DNA amplified at a C_t_ of 32.54, a threshold of 30 C_t_ can be implemented, following the convention of setting an appropriate threshold before the earliest amplifying non-target (e.g., of 2–3 cycles) [26,40]. The low copy number input (7.21 copies/µL) was intended to reflect the typical quantity of DNA that can be extracted from processed material (based on 6 µL DNA extraction elute). Since the *Jacaratia digitata* DNA amplified at a C_t_ of 42.60, a threshold of 40 C_t_ can be implemented when using the assay for DNA extracted from processed botanical material.

These C_t_ cutoff determinations come with an important caveat. Species that are closely related to the target of the assay are useful to include in the exclusivity panel to objectively ensure superior specificity. However, the ultimate judgment of an assay being fit-for-purpose also involves consideration of which non-targets are a real threat to adulteration, and consequently must be avoided as triggering false positives. *Jacaratia digitata* and *Jacaratia dolichaula* have limited commercial relevance in the context of *C. papaya*. In these cases, where there is no economic incentive for adulteration, the C_t_ thresholds that were imposed due to the late off-target amplification may be ignored. We included these two species in our analysis as an important demonstration of the utility of synthetic DNA in specificity testing. In a case where the two off targets were commercially relevant, this gBlock^®^ technique may be the only accessible method by which specificity could be empirically evaluated.

In addition to specificity, superior efficiency, sensitivity, repeatability, and reproducibility are hallmarks of an effective hydrolysis-probe assay [31]. If the intended use of an assay is only qualitative, it is more tolerant to imperfect efficiency, thus an acceptable range lies in between 80–120% [31,32]. However, an efficiency close to 100% is helpful in determining the influence of matrix effects (e.g., influences on target amplification based on presence of PCR inhibitors) on reactions, because poor efficiency can be eliminated as an influencing variable [13]. The *Carica papaya* assay had superior efficiency, evaluated at 104% using USA-origin papaya DNA, and 105% using Brazil-origin papaya DNA. Construction of the standard curves also allowed for determination of an LOD at 25 fg/µL papaya DNA. Based on the copy number determination of ddPCR (28,800 copies/µL = 0.25 ng/µL) this LOD can be expressed as 2.88 copies/µL. This LOD accommodates the typical copy number concentration of DNA extracted from processed papaya powder (7.21 copies/µL). This indicates the ability of the assay to be used for detection of *Carica papaya* DNA in a processed powder matrix.

Lastly, the reliability and reproducibility of the assay provide a metric with which to judge the practicability of using the assay in a laboratory environment. In both series of tests, 100% amplification of true positives and 0% amplification of true negatives were observed. It is only with this consistent performance that the assay can be considered for inclusion in a diagnostic laboratory’s quality-control program. In this validation study, it was demonstrated that the *C. papaya* assay has the ability to detect a specific, target DNA sequence in genomic material extracted from both fresh papaya and industrially processed papaya extract matrices.

### 4.2. Investigating Accuracy Using ddPCR

The primers and probe that were designed for the real-time PCR assay were also used in ddPCR experiments: first, to correlate template input in nanograms of DNA to the copy number of the target sequence, and second, to contrast the interpretation of ddPCR to real-time PCR. In the scenarios depicted in Figure 4, high-input fresh papaya DNA showed a clear positive high-fluorescent signal from all droplets. When comparing the results from the non-target species that amplified the earliest in real-time PCR (*Jacaratia digitata* DNA) and amplification of DNA from processed papaya powder, there was a clear fluorescent intensity separation between true positives and non-target DNA. Despite the low-quantity DNA input from the processed papaya powder sample, a clear cluster of high-fluorescent droplets appears at >800 amplitude. Though *Jacaratia digitata* DNA does interact with the assay, the amplification is inefficient, revealing a spread of droplets from mid- (<800) to low-fluorescence. Whereas a C_t_ cutoff must be implemented in real-time PCR, in order not to confuse late amplification of this non-target with a true positive, ddPCR results allow for a more objective determination of positives.

## 5. Conclusions

This manuscript outlined the successful design and validation of a hydrolysis-probe-based real-time PCR assay for the detection of *Carica papaya* DNA. Performance of the assay was validated according to guidelines described in current literature, centered around evaluation of common parameters: specificity, efficiency, sensitivity, repeatability, and reproducibility. Given the paramount importance of specificity in the development of a botanical detection assay, we discussed the importance of constructing a comprehensive exclusivity panel and proposed a viable option for completing a panel when botanical samples are difficult to procure. Use of synthetic DNA (in this case a gBlock^®^), that is representative of the homolog to the target sequence of the target species, allows for empirical evaluation of off-target amplification. This technique can be easily implemented by firstly using ddPCR to determine the target sequence copy-number in genomic DNA, to which synthetic DNA copy-number should be corrected. The necessary inclusion of this specificity evaluation (which revealed late amplification of non-targets) allowed for appropriate C_t_ cutoffs to be set, thus determining LOD and ensuring accuracy in assay performance. Through empirical validation of performance metrics, we developed an assay that can be appropriately used for the detection of *Carica papaya* DNA in fresh and processed fruit matrices.

## Figures and Tables

**Figure 1 foods-12-00530-f001:**
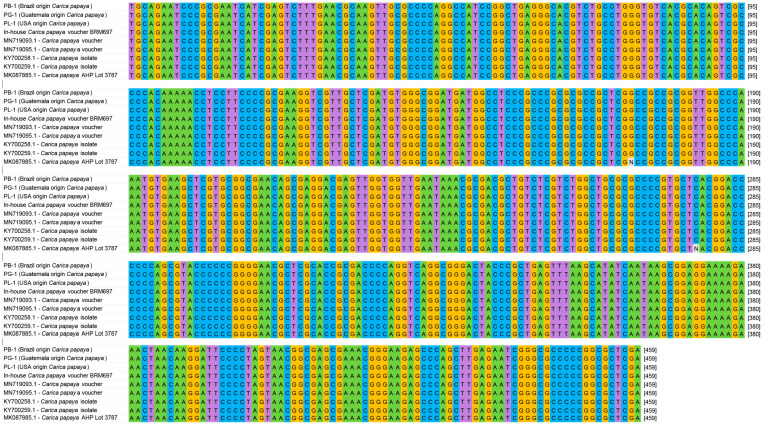
Alignment of ITS2 sequences from fresh fruit papaya samples used in this study to voucher and reference material sequences collected in-house and from public databases.

**Figure 2 foods-12-00530-f002:**
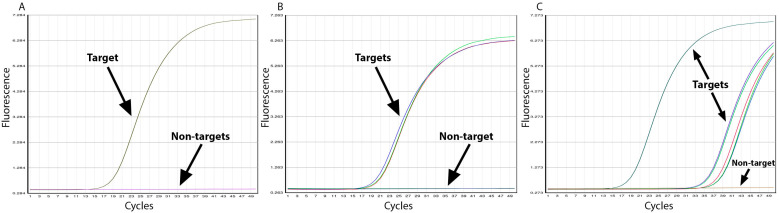
Evaluation of analytical specificity of the *C. papaya* assay. Graphs show: (**A**) positive amplification of one target papaya BRM (3787) and 21 unamplified non-target species, (**B**) positive amplification of three target fresh papaya samples (PB–1, PG–1, PL–1) and one unamplified negative control non-target, and (**C**) positive amplification of one fresh papaya sample (BRM697), five processed papaya powder lots (Batch R12418: lot 1036836, 1036961, 1039115, and Batch R0003: lot 1047426, 1050403) and one unamplified negative control non-target.

**Figure 3 foods-12-00530-f003:**
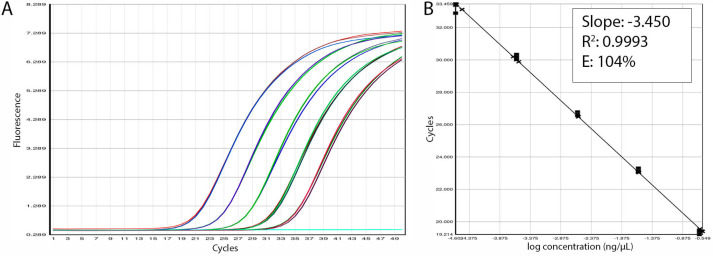
(**A**) A standard curve created using serial dilutions of DNA extracted from USA-sourced *Carica papaya*. (**B**) Linearity and efficiency calculations from the standard curve.

**Figure 4 foods-12-00530-f004:**
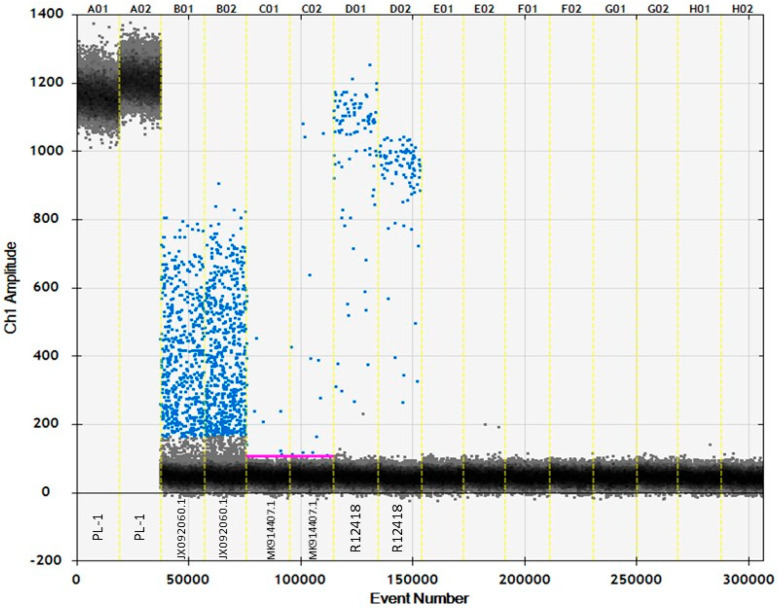
ddPCR results of fresh papaya (PL–1), processed papaya extract (R12418), *Jacaratia dolichaula* (JX092060.1), and *Jacaratia digitata* (MK914407.1) DNA, in duplicate, based on 40 PCR cycles. The purple line (threshold between positive and negative droplets) is marked where insufficient positive droplets resulted in a “negative” overall sample call.

**Table 1 foods-12-00530-t001:** Target and non-target samples used for validation of *Carica papaya* assay.

Botanical Name	Family	Code	Source	Common Name	Type of Sample	Type of Material
*Carica papaya*	Caricaceae	PB–1 (Brazil)	Market	Papaya	Target	Fruit
*Carica papaya*	Caricaceae	PG–1 (Guatemala)	Market	Papaya	Target	Fruit
*Carica papaya*	Caricaceae	PL–1 (USA)	Market	Papaya	Target	Fruit
*Carica papaya*	Caricaceae	R12418 *	Martin Bauer	Papaya	Target	8–10% fruit extract powder
*Carica papaya*	Caricaceae	R00003 **	Martin Bauer	Papaya	Target	8–10% fruit extract powder
*Carica papaya*	Caricaceae	3787	AHP	Papaya	Target	Dried fruit
*Carica papaya*	Caricaceae	BRM697	In-house voucher	Papaya	Target	Dried leaf
*Angelica sinensis*	Apiaceae	SA09609CR38	Alkemist	Angelica	Non-Target	Dried root
*Panax ginseng*	Araliaceae	19091QSS	Alkemist	Asian ginseng	Non-Target	Dried root
*Beta vulgaris*	Amaranthaceae	4413	AHP	Beet	Non-Target	Dried root
*Vaccinium corymbosum*	Ericaceae	5314	Chromadex	Blueberry	Non-Target	Dried fruit
*Matricaria chamomilla L*	Asteraceae	00030692–495	Chromadex	Chamomile	Non-Target	Dried flower
*Mentha canadensis* syn. *haplocalyx*	Lamiaceae	30984–241	Chromadex	Chinese mint	Non-Target	Dried leaf
*Zea mays*	Poaceae	00031127–356	Chromadex	Corn silk	Non-Target	Dried stigma
*Taraxacum officinale*	Asteraceae	00030662–697	Chromadex	Dandelion	Non-Target	Dried root
*Rosa canina*	Rosaceae	00030792–473	Chromadex	Dog rose	Non-Target	Dried fruit
*Zingiber officinale*	Zingiberaceae	5374	AHP	Ginger	Non-Target	Dried root
*Camellia sinensis*	Theaceae	00030330–054	Chromadex	Green tea	Non-Target	Dried leaf
*Paullinia cupana*	Sapindaceae	00030335–064	Chromadex	Guarana	Non-Target	Dried seed
*Melissa officinalis*	Lamiaceae	3335.6	AHP	Lemon balm	Non-Target	Dried herb
*Ganoderma lucidum*	Ganodermataceae	H20109CRB10	Alkemist	Lingzhi Mushroom	Non-Target	Dried, whole mushroom
*Allium cepa*	Amaryllidaceae	4533	AHP	Onion	Non-Target	Dried bulb
*Rosmarinus officinalis*	Lamiaceae	5063	AHP	Rosemary	Non-Target	Dried leaf
*Schisandra chinensis*	Schisandraceae	3241.4	AHP	Schisandra	Non-Target	Dried fruit
*Spinacia oleracea*	Amaranthaceae	4647	Chromadex	Spinach	Non-Target	Dried leaf
*Curcuma longa*	Zingiberaceae	00031107–328	Chromadex	Turmeric	Non-Target	Dried root
*Daucus carota*	Apiaceae	00031080–288	Alkemist	Wild carrot	Non-Target	Dried root
*Petroselinum crispum*	Apiaceae	UR29409CRB15	AHP	Parsley	Non-Target	Dried leaf
*Jacaratia dolichaula*	Caricaceae	JX092060.1	IDT	Barrilillo	Non-Target	gBlock^®^
*Jacaratia digitata*	Caricaceae	MK914407.1	IDT	Papaya Caspi	Non-Target	gBlock^®^

* Three lots were sampled from raw material source R12418: 1036836, 1036961, 1039115; ** Three lots were sampled from raw material source R00003: 1047426, 1047427, 1050403.

**Table 2 foods-12-00530-t002:** Primer and probe sequences used in *Carica papaya* assay.

Oligo	Sequence
Forward Primer	5′-TCG AGT CTT TGA ACG CAA GTT-3′
Reverse Primer	5′-GGG GAA GGA GGT TTT TGT G-3′
Probe	5′-/56-FAM/ACT GTG CGT/ZEN/GAC ACC CAG GCA GA/3IABkFQ/-3′

## Data Availability

Sequence data used in this study is openly available in GenBank. Herbalife reference sequences appear in BioProject: PRJNA503738, and other individual public data accessions are noted in Figure 1 and Appendix A.

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
