# Peer review of "Validation of a Real-Time PCR Assay for Identification of Fresh and Processed Carica papaya Botanical Material: Using Synthetic DNA to Supplement Specificity Evaluation"

_foods, 2023, doi:10.3390/foods12030530_

Round 1

Reviewer 1 Report

Authors describe in their manuscript the Validation of a real-time PCR assay for identification of botanical material, either fresh or processed papaya, using synthetic DNA to supplement specificity evaluation.

The goal of the research has been focused and described starting from the title, the topic can be of interest of the reader and is another brick in the specific researching field.

The results have been critically discussed and the conclusions are consistent with the experimental evidence.

No comments on tables and figures.

The work is well structured, the experimental design is rigorous and the authors critically discuss the results.

I suggest checking whether any recent bibliography has been omitted

Author Response

Note: our response appears below your comments

The goal of the research has been focused and described starting from the title, the topic can be of interest of the reader and is another brick in the specific researching field. The results have been critically discussed and the conclusions are consistent with the experimental evidence. No comments on tables and figures. The work is well structured, the experimental design is rigorous and the authors critically discuss the results. I suggest checking whether any recent bibliography has been omitted.

Thank you for taking the time to review our manuscript and provide comments. We have taken your suggestion to revisit our references list and we have conducted further literature review and added several more references that better describe the background information of the work (e.g., a more comprehensive review of previous efforts to design fingerprinting methods for papaya identification). We have also edited sections of the abstract, introduction, results, and discussion to improve clarity and emphasize the impact of the work.

Reviewer 2 Report

The article entitled Validation of a real-time PCR assay for identification of fresh and processed Carica papaya botanical material: Using synthetic DNA to supplement specificity evaluation is an interesting one but I have some concerns regarding it.

The abstract must be improved there are some lacks of writing in the first sentences.

It would be interesting to submit your results to a statistical analysis such as PCA to see how the samples are gathered in function of their authenticity.

The discussion section must be improved.

Author Response

Note: our response appears below your comments

The article entitled Validation of a real-time PCR assay for identification of fresh and processed Carica papaya botanical material: Using synthetic DNA to supplement specificity evaluation is an interesting one but I have some concerns regarding it.

Thank you for taking the time to review our work and make suggestions. We have made several edits and added references and information to better clarify the approach, outcomes, and impact of our research.

The abstract must be improved there are some lacks of writing in the first sentences. It would be interesting to submit your results to a statistical analysis such as PCA to see how the samples are gathered in function of their authenticity. The discussion section must be improved.

We have taken your suggestion and edited the abstract to better communicate the intended focus and impact of the work. I clarified our goal of this study serving as a guideline for other validation studies which may otherwise have skipped difficult-to-procure species in their exclusivity panels. We hope that they will consider supplementing the exclusivity panel with non-targets represented by synthetic DNA, based on the method we outlined. Further, while we did not believe that a PCA analysis would be appropriate for our data, we made several edits throughout the manuscript to better communicate the reasoning behind the investigation. We added several references in the introduction and discussion to better describe background research and justify exploration of the method that we present.

We made several improvements as summarized below:

-Introduction: Upon further literature review, I added mention of papaya production increase being driven by expanded cultivation (allowed by advanced growing practices like greenhouses in Mediterranean regions). I also included further discussion and examples of previous investigation into and development of fingerprinting methods for papaya identification (both genomic and chemical approaches). This helps set up an argument for the justification of exploring the method we present in our study.

-Methods: I changed presentation of the sequences to include them in a table as opposed to the text (Table 2).

-Results: I added two more panels to Figure 1 (specificity evaluation) to show amplification of key known positive samples that I had listed in Table 1. I added two tables (Supplementary Table S1 and S2) that summarize the data that are described in section 3.4.

-Introduction and Discussion: I completed further literature review and added several informative, recent citations that support the background research of this work. I edited the discussion to improve clarity and highlight the impact of our research.

Thank you again for your valued comments and efforts to improve this work.

Reviewer 3 Report

The manuscript titled " Validation of a real-time PCR assay for identification of fresh and processed Carica papaya botanical material: Using synthetic DNA to supplement specificity evaluation," authored by Patel and colleagues, aims to validate a PCR protocol to identify fresh or processed Carica papaya products.

The manuscript is excellently written, using a propriety of language typical of professionals and experts in the field. Moreover, it contains very interesting data that seriously can contribute to the knowledge of the field and counteract processes of sophistication and adulteration typical of the food and dietary supplement industry. Indeed, the method proposed by the authors has considerable and varied applications.

Below are just a number of observations:

AFFILIATION SECTION: Authors should report in the affiliations section the email address along with the acronym for each author. This same acronym should be the one used in the contributions section.

ABSTRACT: In general this section is fine. However, one shortcoming of the abstract section is the absence of a conclusion sentence highlighting the practical application of the authors' research and future prospects.

KEYWORDS: some keywords should be changed. The utility of these terms is to facilitate the search of the article using common scientific search engines (PubMed, GoogleScholar, Scopus, etc.), which rely on the terms contained in title, abstract, and keywords. Consequently, using terms that are already in these sections as keywords is inappropriate. I strongly suggest that the repetitive keywords be changed before re-submission.

INTRODUCTION: in the first paragraph of this section, the authors should mention that one of the reasons why the Papaya market will grow exponentially over the years is also due to climate change. Indeed, while previously the cultivation of this plant was exclusively peculiar to sub-tropical and tropical regions, it now sees suitable conditions for the growth and ripening of the fruit also in Mediterranean area, such as Sicily, Calabria, and Spain, which due to (or thanks to) climate change have developed proper conditions for a suitable cultivation of sub-tropical species (REF: doi.org/10.3390/agronomy10040501). Moreover, fingerprinting techniques for plant matrix identification, are not exclusively limited to HPLC and DNA-based methods. Indeed, fingerprinting based on minerals, volatile compounds, microbiome, or other types of markers are also currently used for plant identification. In addition, the authors should mention that fingerprinting techniques are also used for identifying the geographic origin of a particular plant. This last point does not emerge completely in the section.

MATERIALS AND METHODS: this section is fairly well described, however, the main problem is the lack of bibliographic references to justify the use of these methodologies. Strongly suggest that the entire section be supplemented with literature references describing the methodologies used. Also, in section 2.4, why were these primers chosen? Is there a previous bibliographic reference? Also, it would be appropriate for the primers to be listed in a table. In section 2.5. the equations should be reported as described in the guidelines of the journal.

RESULTS: the authors correctly describe the specificity of their assay, however, they do not show the results obtained from the samples shown in Table 1. The authors are strongly encouraged to integrate these data into the main text of the manuscript. In particular, they could create a single figure consisting of several panels, trying to group the samples by type. Moreover, also data described in 3.4. subsection should be reported at least as supplementary data.

REFERENCES: I strongly suggest that the article references be implemented, even considering the large number of self-references. This reviewer is not asking for the removal of self-citactions, as I believe they are quite relevant to the purpose of the manuscript. However, the limited number of total references excessively increases the self-citation rate.

Author Response

Note: our responses appear below your comments.

The manuscript is excellently written, using a propriety of language typical of professionals and experts in the field. Moreover, it contains very interesting data that seriously can contribute to the knowledge of the field and counteract processes of sophistication and adulteration typical of the food and dietary supplement industry. Indeed, the method proposed by the authors has considerable and varied applications.

Thank you very much for your helpful and detailed comments. We sought to capture your suggestions in our revised manuscript as described below:

Below are just a number of observations:

AFFILIATION SECTION: Authors should report in the affiliations section the email address along with the acronym for each author. This same acronym should be the one used in the contributions section.

The corresponding author’s email is included in the affiliations section, and we will consult the journal editors to see if we can include the acronyms in this section as well, based on their design guidelines.

ABSTRACT: In general this section is fine. However, one shortcoming of the abstract section is the absence of a conclusion sentence highlighting the practical application of the authors' research and future prospects.

I have edited the abstract to better communicate the intended focus and impact of the work. I clarified our goal of this study serving as a guideline for other validation studies which may otherwise have skipped difficult-to-procure species in their exclusivity panels. We hope that they will consider supplementing the exclusivity panel with non-targets represented by synthetic DNA, based on the method we outlined.

KEYWORDS: some keywords should be changed. The utility of these terms is to facilitate the search of the article using common scientific search engines (PubMed, GoogleScholar, Scopus, etc.), which rely on the terms contained in title, abstract, and keywords. Consequently, using terms that are already in these sections as keywords is inappropriate. I strongly suggest that the repetitive keywords be changed before re-submission.

I have reworked the list of keywords.

INTRODUCTION: in the first paragraph of this section, the authors should mention that one of the reasons why the Papaya market will grow exponentially over the years is also due to climate change. Indeed, while previously the cultivation of this plant was exclusively peculiar to sub-tropical and tropical regions, it now sees suitable conditions for the growth and ripening of the fruit also in Mediterranean area, such as Sicily, Calabria, and Spain, which due to (or thanks to) climate change have developed proper conditions for a suitable cultivation of sub-tropical species (REF: doi.org/10.3390/agronomy10040501). Moreover, fingerprinting techniques for plant matrix identification, are not exclusively limited to HPLC and DNA-based methods. Indeed, fingerprinting based on minerals, volatile compounds, microbiome, or other types of markers are also currently used for plant identification. In addition, the authors should mention that fingerprinting techniques are also used for identifying the geographic origin of a particular plant. This last point does not emerge completely in the section.

Thank you for the suggested reference. I added mention of papaya production increase being driven by expanded cultivation (allowed by climate change and advanced growing practices like greenhouses in Mediterranean regions). I also included further discussion and examples of previous investigation into and development of fingerprinting methods for papaya identification (both genomic and chemical approaches). This helps set up an argument for the justification of exploring the method we present in our study.

MATERIALS AND METHODS: this section is fairly well described, however, the main problem is the lack of bibliographic references to justify the use of these methodologies. Strongly suggest that the entire section be supplemented with literature references describing the methodologies used. Also, in section 2.4, why were these primers chosen? Is there a previous bibliographic reference? Also, it would be appropriate for the primers to be listed in a table. In section 2.5. the equations should be reported as described in the guidelines of the journal.

I supplemented existing references with others (e.g., Bohme et al., 2019) that discuss the advantages of real-time PCR approaches to botanical identification, in order to help justify the work (though I believe that some of these references were better suited for the introduction, so I placed them there). There are four citations in section 2.2 of the methods section that represent the reasoning and decisions behind our methodologies well.

I added a sentence in section 2.4 clarifying that the primers were designed and chosen based on their in silico specificity evaluation. The design of primers/probe is discussed in more detail in section 2.2. I also changed presentation of the sequences to include them in a table as opposed to the text (Table 2).

I reformatted the equations in section 2.5 to fit the template of the journal.

RESULTS: the authors correctly describe the specificity of their assay, however, they do not show the results obtained from the samples shown in Table 1. The authors are strongly encouraged to integrate these data into the main text of the manuscript. In particular, they could create a single figure consisting of several panels, trying to group the samples by type. Moreover, also data described in 3.4. subsection should be reported at least as supplementary data.

As per your suggestion, I added two more panels to Figure 1 (specificity evaluation) to show amplification of key known positive samples that I had listed in Table 1. I added two tables (Supplementary Table S1 and S2) that summarize the data that are described in section 3.4.

REFERENCES: I strongly suggest that the article references be implemented, even considering the large number of self-references. This reviewer is not asking for the removal of self-citactions, as I believe they are quite relevant to the purpose of the manuscript. However, the limited number of total references excessively increases the self-citation rate.

I completed further literature review and added several informative, recent citations that support the background research of this work. Further, I removed self-citations if there was an alternative that effectively communicated the same message.

Thank you again for your detailed comments and valued contribution to improving this work.

Round 2

Reviewer 2 Report

Accept as it is.